# MDPO: Overcoming the Training-Inference Divide of Masked Diffusion Language Models

## Abstract

Diffusion language models, as a promising alternative to traditional autoregressive (AR) models, enable faster generation and richer conditioning on bidirectional context. However, they suffer from a key discrepancy between training and inference: during inference, MDLMs progressively reveal the structure of the generated sequence by producing fewer and fewer masked tokens, whereas this structure is ignored in training as tokens are masked at random. Although this discrepancy between training and inference can lead to suboptimal performance, it has been largely overlooked by previous works, leaving closing this gap between the two stages an open problem. To address this, we frame the problem of learning effective denoising trajectories as a sequential decision-making problem and use the resulting framework to apply reinforcement learning. We propose a novel Masked Diffusion Policy Optimization (MDPO) to exploit the Markov property diffusion possesses and explicitly train the model under the same progressive refining schedule used at inference. MDPO matches the performance of the previous state-of-the-art (SOTA) method with 60× fewer gradient updates, while achieving average improvements of 9.6% on MATH500 and 54.2% on Countdown over SOTA when trained within the same number of weight updates. Additionally, we improve the remasking strategy of MDLMs as a plug-in inference replacement to overcome the limitation that the model cannot refine tokens flexibly. This training-free method, termed Running Confidence Remasking (RCR), consistently enhances performance and provides further improvements when used with MDPO. Our findings establish great potential for investigating the discrepancy between pre-training and inference of MDLMs. To support reproducibility, we will release code upon acceptance.

## 1 Introduction

Remarkable advances in language modeling have largely been driven by the autoregressive (AR) paradigm (Hochreiter & Schmidhuber, 1997; Vaswani et al., 2017), where tokens are generated causally. Diffusion Language Models (DLMs), as an alternative line of work, generate by iterative denoising to enable parallel prediction, bidirectional conditioning, and faster generation speed. Recently, DLMs such as Mercury Coder (Khanna et al., 2025), Google Gemini Diffusion (Google DeepMind, 2025), and Seed Diffusion (Song et al., 2025) have emerged as promising alternatives that excel at external math and coding benchmarks. Built upon theoretical insights and practical optimizations in the discrete diffusion framework (Lou et al., 2024; Sahoo et al., 2024; Ou et al., 2025), Masked Diffusion Language Models (MDLMs) lead the current development of DLMs. In the forward process of MDLMs, tokens are randomly chosen to either remain stationary or transition to a predefined absorbing state, such as a special masking token, motivated by the success of BERT (Devlin et al., 2019). Then, in the backward process, the model learns to invert the noise by predicting *all* masked tokens conditioned on partially observed (unmasked) context. For generating a sequence of tokens, current MDLMs alternate between predicting masked tokens and selectively remasking a fraction of the predictions based on heuristics and uncalibrated token-wise scores. Representative works such as LLaDA-8B (Nie et al., 2025) and Dream-7B (Ye et al., 2025) demonstrate competitive performance to their similarly sized AR counterparts.

Despite promising empirical results, we identify two fundamental, largely overlooked problems of MDLMs that hinder effective denoising trajectories. (1) The *training–inference divide*: at inference

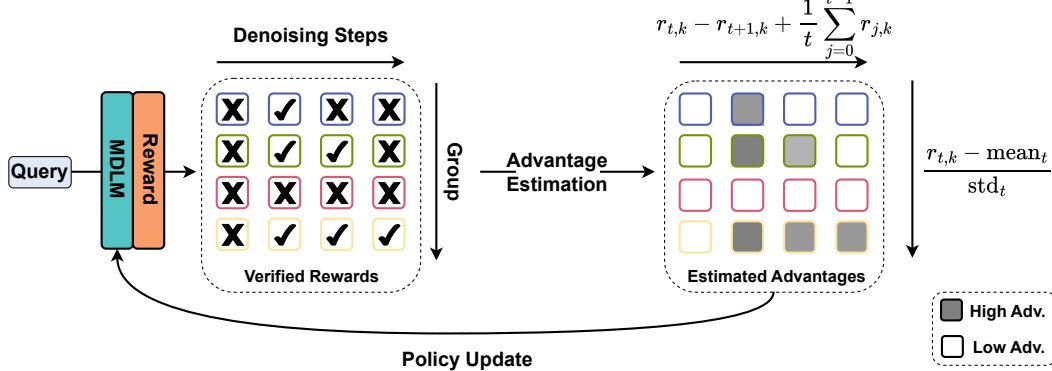

Figure 1: **MDPO** generates a group of answers given a query for RL rollouts. Then all completions at intermediate and final steps are verified with a reward model. Based on verified rewards, MDPO estimates the advantage of step $t$ by considering rewards of the other steps in the current rollout and step $t$ from other rollouts in the group. These estimated advantages are used for policy optimization.

time, MDLMs follow a model-dependent, confidence-guided remasking schedule that progressively reveals the structure of the generated sequence. By contrast, tokens are randomly masked at sampled diffusion steps in training, ignoring the progressive schedule. This mismatch prevents the model from learning effective denoising trajectories and often causes over-denoising: correct intermediate solutions are 'refined' into wrong final solutions (see Fig. 3 for statistical analysis). (2) The common remasking approaches freeze tokens once predicted, making it impossible to revise early low-confidence predictions at later denoising steps. We address these problems with complementary methods in this work. Masked Diffusion Policy Optimization (MDPO) casts denoising as a multi-step decision-making process, explicitly training the model under the same progressive, model-dependent remasking schedule used at inference. By using reinforcement learning with intermediate rewards to optimize these trajectories, MDPO aligns training with actual inference dynamics, closing the training–inference divide. Running Confidence Remasking (RCR) is a training-free decoding strategy that tracks per-position confidence over time, enabling flexible remasking and revision of low-confidence tokens at any step. In summary, our contributions are threefold:

- **Over-denoising:** We observe a novel and previously unknown phenomenon in MDLMs: models occasionally produce correct answers at intermediate steps but 'refine' them into incorrect results. We refer to this phenomenon as over-denoising, which inspires us to supervise models not only on the final results but also intermediate denoising steps.
- **Masked Diffusion Policy Optimization (MDPO):** We propose MDPO to learn effective denoising trajectories without direct supervision. By exploiting the fact that MDLMs yield complete generations at every inference step, MDPO optimizes the model with policy gradients via intermediate-step rewards. Unlike prior RL fine-tuning on MDLMs, MDPO explicitly targets addressing the *training-inference divide* overlooked by previous works.
- **Running Confidence Remasking (RCR):** Instead of freezing tokens based on their single-step confidence, RCR allows flexible remasking by continuously tracking the running lowest confidence over denoising trajectories. Our experiments demonstrate consistently improved performance of using RCR on both LLaDA pre-trained as well as MDPO fine-tuned models.

Empirical results on challenging mathematical and reasoning benchmarks show that both MDPO and the training-free RCR substantially improve generation quality and sample efficiency. Our findings underscore the value of addressing the training-inference divide for masked diffusion language models and open new directions for research in this emerging paradigm.

## 2 METHODOLOGY

### 2.1 PRELIMINARY: MASKED DIFFUSION LANGUAGE MODELS

We introduce our methods based on the formulation of LLaDA (Nie et al., 2025), a pre-trained MDLM that models clean data $x_0$ via a discrete diffusion process (Ou et al., 2025; Austin et al.,

2021), where tokens are progressively masked in the forward process and then recovered by iterative prediction in the reverse process. While specific methodological variations may exist, the methods proposed in this paper are broadly applicable to the majority of existing MDLM frameworks.

**Training:** The training objective optimizes a mask predictor $p_\theta (x_0 \mid \bar{x}_\tau)$ that takes partially masked data $\bar{x}_\tau$[1] as input and predicts all masked tokens simultaneously using a cross-entropy loss:

$$\mathcal{L}(\theta) \triangleq -\mathbb{E}_{x_0, \tau} \left[ \frac{1}{\tau} \sum_{i=1}^{L} \mathbb{1}[\bar{x}_\tau^i = \mathcal{M}] \, \log p_\theta \left(x_0^i \mid \bar{x}_\tau\right) \right] \qquad \bar{x}_\tau \triangleq \tilde{\gamma}(x_0, \tau) \qquad (1)$$

Here, the expectation is computed over the unmasked ground truth sequences $\{x_0\}$ in the dataset and a randomly drawn masking ratio $\tau \sim U(0,1)$. $\tilde{\gamma}(x_0, \tau)$ is a masking function that sets each token of $x_0$ to the mask token $\mathcal{M}$ independently with probability $\tau$. $L$ is the sequence length and superscript $i$ for both $x$ and $\bar{x}$ denotes the $i$'th element in the sequence. The cross-entropy loss is only computed on the masked tokens as indicated by the function $\mathbb{1}[\cdot]$. In practice, $p_\theta (x_0 \mid \bar{x}_\tau)$ often depends on an additional input prompt which we drop here for notational clarity.

**Inference:** Inference of LLaDA involves iterative refinement from a fully masked input, progressively reducing the number of masked tokens at each step $t$. This is necessary, as the initial noise contains no structure and the mapping from noise to the generated sequence is highly non-linear and multimodal (Xiao et al., 2022; Song et al., 2021). Hence, multiple iterations (each conditioning on the structure generated so far) are required to achieve coherent, contextually and syntactically consistent outputs (Nie et al., 2025; Feng et al., 2025). Given $T$ denoising steps, the inference starts from a fully masked sequence $\bar{x}_T = (\mathcal{M}, \ldots, \mathcal{M})$ and alternates the following two steps:

1. Given a masked sequence $\bar{x}_t$, draw all masked tokens by sampling $x_{t-1} \sim p_\theta(\cdot \mid \bar{x}_t)$ to produce a *fully unmasked* sequence $x_{t-1}$.

2. Remask $x_{t-1}$ using a confidence-based remasking function $\gamma$ yielding a *partially masked* sequence $\bar{x}_{t-1}$. More specifically, with decreasing $t$, LLaDA remasks a decreasing number of tokens based on the current model's confidence score $p_\theta(x_{t-1}^i \mid \bar{x}_t)$

LLaDA continues this process until $x_0$ is sampled. The confidence-based remasking strategy in the second step above reveals more and more structure for $p_\theta(\cdot \mid \bar{x}_t)$ in the first step to condition on over time. $T$ is a hyperparameter that trades off computation and quality (Nie et al., 2025) in practice. We refer readers to Section 2.4 for details about the implementation of the remasking function $\gamma$.

## 2.2 THE TRAINING-INFERENCE DIVIDE

The above formulation of LLaDA reveals an important discrepancy between training and inference: During training, the model is optimized to predict all masked tokens at randomly sampled masking ratios $\tau$. In contrast, inference involves multiple iterative denoising steps informed by the model's confidence score, forming a trajectory with fewer and fewer masked tokens to be predicted, and more and more structure being revealed. However, this structure is ignored during training where tokens are masked at random at each iteration. This *training-inference divide* limits the performance of the model as the model is unable to learn effective denoising trajectories.

We observe symptoms of this divide by inspecting denoising trajectories of pre-trained MDLMs. Table 1 shows the accuracy of LLaDA on two reasoning tasks and the additional accuracy if we count intermediate-step correct answers that are 'refined' into wrong final answers. We term this phenomenon as over-denoising and conduct detailed studies in Section 3.3.

A simple solution to address this discrepancy would be to fine-tune the model with ground-truth trajectories. However, such trajectories are inherently unavailable, as human-generated data does not capture iterative denoising paths. In this paper, we hence investigate the following question:

| Task | Acc. |
|---|---|
| MATH-500 | 18.2 |
| w/ inter. ans | +9.8 |
| Countdown | 38.7 |
| w/ inter. ans | +6.2 |

Table 1: Accuracy of LLaDA on two verifiable tasks and the ratio of at least one out of 256 intermediate steps is correct (w/ inter.ans).

---

[1]We use the bar notation $\bar{x}$ to denote (partially) masked and $x$ for unmasked sequences throughout the paper.

> How can we learn effective discrete diffusion trajectories without direct supervision?

A key advantage of masked diffusion language models over traditional auto-regressive models is that they yield *complete* text generations at *every* inference step. This allows for evaluating the quality of intermediate generations using an appropriately chosen reward model. In this paper, we propose to exploit this property to effectively fine-tune diffusion policies using deep reinforcement learning. To focus our analysis, we investigate reasoning tasks with verifiable answers, such as math problems. However, we remark that other problems without verifiable answers can also be addressed within our framework using modern validation concepts such as LLM-as-a-judge (Li et al., 2024).

## 2.3 MASKED DIFFUSION POLICY OPTIMIZATION (MDPO)

While reinforcement learning (RL) has originally been applied to address sequential decision-making tasks, it emerges as a promising alternative for optimizing deep neural networks to maximize non-differentiable objectives (Mnih et al., 2013; François-Lavet et al., 2018; Ouyang et al., 2022). Recognizing inference in MDLMs as a sequential decision-making problem, we propose to explicitly train the mask prediction network as an RL policy (Jaeger & Geiger, 2024). Policy gradient methods train a policy network $\pi(a \mid s)$ that predicts a probability distribution over actions using non-differentiable rewards by maximizing the expected return:

$$\mathcal{J}(\pi) = \mathbb{E}_{(s,a)} \left[ \pi(a \mid s) \, r(s,a) \right] \tag{2}$$

where the expectation is computed over state-action pairs $(s, a)$ collected on-policy and $r(s, a)$ is a reward function. Linking MDLMs to policy gradient methods, each denoising step corresponds to an action that predicts all masked tokens in a (partially) masked sequence $\bar{x}_t$, with reward $r(x_{t-1})$ via an evaluation model $r(\cdot)$.

Based on this, we derive the expected return of our Masked Diffusion Policy Optimization:

$$\mathcal{J}(\theta) \triangleq \mathbb{E}_{\{x_T,..,x_0\} \sim p_\theta} \left[ \sum_{t=1}^{T} \sum_{i=1}^{L} \mathbb{1}[\bar{x}_t^i = \mathcal{M}] \, \log p_\theta(x_{t-1}^i \mid \bar{x}_t) \, r(x_{t-1}) \right] \qquad \bar{x}_t \triangleq \gamma(x_t, t) \tag{3}$$

where the expectation is computed over denoising trajectories $\{x_T, \dots, x_0\}$ generated by the current denoising policy $p_\theta$. We use the same masking function $\gamma(x_t, t)$ as used during inference, which takes an unmasked sequence $x_t$ as input and converts it into a (partially) masked sequence $\bar{x}_t$ conditioned on the current time step $t$. The smaller the timestep, the fewer tokens are masked (see Section 2.4 for details). The indicator function aggregates the return only at locations where the input is masked. The denoising policy models the probability of the denoised token $x_{t-1}^i$ at location $i$ given the masked input at the previous time step $\bar{x}_t$. Again, we drop the prompt here for notational clarity.

As the gradients must be computed from data generated by the current policy $p_\theta$, the policy gradient estimator defined in Eq. (3) only allows a single optimization step for each round of data collection. To enable multiple steps of optimization, we use importance sampling (Kakade & Langford, 2002; Black et al., 2024) and the clipped surrogate objective from Proximal Policy Optimization (PPO) (Schulman et al., 2017). Moreover, inspired by the effectiveness of group-relative advantage estimation (Shao et al., 2024), we sample a group of $G$ trajectories $\{x_{t,1}, \dots, x_{t,G}\}$ at each step $t$ from the old policy $p_{\theta_{\text{old}}}$ to update the current policy $p_\theta$, see Fig. 1 for demonstration. Our final objective is given by:

$$\mathcal{J}(\theta) \triangleq \mathbb{E}_{\{x_{T,k}, \dots, x_{0,k}\}_{k=1}^{G} \sim p_{\theta_{\text{old}}}} \left[ \frac{1}{G} \sum_{k=1}^{G} \sum_{t=1}^{T} \sum_{i=1}^{L} \mathbb{1}[\bar{x}_{t,k}^i = \mathcal{M}] \left( Z_{t,k}^i - \beta \mathbb{D}_{KL} \left[ p_\theta || p_{\text{ref}} \right] \right) \right] \tag{4}$$

Here, $p_{\text{ref}}$ is the reference policy which is usually the initial model before training. $\mathbb{D}_{KL} \left[ p_\theta || p_{\text{ref}} \right]$ is the KL divergence between the trained policy $p_\theta$ and the reference policy to avoid large deviation. We define the clipped and weighted advantage (Schulman et al., 2017) as

$$Z_{t,k}^i = \min \left[ \frac{p_\theta(x_{t-1,k}^i \mid \bar{x}_{t,k})}{p_{\theta_{\text{old}}}(x_{t-1,k}^i \mid \bar{x}_{t,k})} A_{t-1,k}, \text{clip} \left( \frac{p_\theta(x_{t-1,k}^i \mid \bar{x}_{t,k})}{p_{\theta_{\text{old}}}(x_{t-1,k}^i \mid \bar{x}_{t,k})}, 1-\epsilon, 1+\epsilon \right) A_{t-1,k} \right] \tag{5}$$

where $\epsilon$ and $\beta$ are hyper-parameters. $A_{t,k}$ is the advantage of $x_{t,k}$, the $k$'th rollout in the group, calculated based on relative rewards inside the group:

$$A_{t,k} = \frac{s_{t,k} - \text{mean}(\{s_{t,1}, s_{t,2}, \cdots, s_{t,G}\})}{\text{std}(\{s_{t,1}, s_{t,2}, \cdots, s_{t,G}\})} \qquad s_{t,k} = r(x_{t,k}) - r(x_{t+1,k}) + \frac{1}{t} \sum_{j=0}^{t-1} r(x_{j,k}) \tag{6}$$

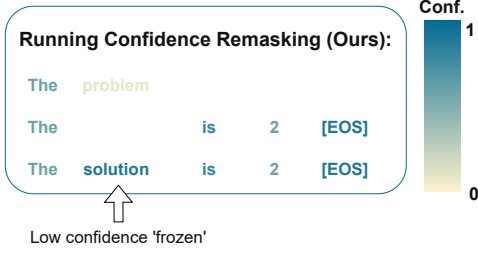

Figure 2: Comparison between Low-Confidence Remasking (LCR) and our proposed Running Confidence Remasking (RCR) during iterative denoising. For each step, we show tokens that are *not* remasked. LCR freezes low-confidence tokens once unmasked, preventing further refinement, which potentially accumulates early-stage noise. For example, the token 'problem' predicted and frozen by LCR at step 1 is wrong but maintained until the end of the denoising, which leads to the final wrong answer. Whereas RCR tracks the running maximum confidence for each position, allowing persistently low-confidence tokens to be refined in later steps, leading to higher-quality completions.

The original group-relative advantage estimation optimizes the memory usage of training LLMs with RL by dropping the jointly trained value model that is used to predict the advantages. Beyond this, another intuitive reason to use group-relative advantage estimation in the diffusion setting is to incentivize better generation with fewer denoising steps. Specifically, by comparing to a group of trajectories, steps reaching the final reward more rapidly are assigned higher advantages. In addition, Eq. (6) encourages each denoising step to (i) make an immediate improvement over the previous, noisier prediction and (ii) steer the trajectory toward cleaner states that score well overall.

## 2.4 REMASKING

As introduced in Section 2.1, inference in LLaDA alternates between predicting all masked tokens in $\bar{x}_t$ and selectively remasking a fraction of the prediction $x_{t-1}$ via $\bar{x}_{t-1} = \gamma(x_{t-1}, t)$ for iterative refinement. We first discuss random and confidence-based remasking as introduced in LLaDA. Next, we propose a simple yet effective improvement to address a key limitation of confidence-based remasking, which consistently leads to better performance for both LLaDA pre-trained as well as MDPO fine-tuned models. Formally, the remasking function $\gamma$ assigns the mask symbol $\mathcal{M}$ to the $n_t$ tokens with the highest masking scores:

$$\gamma(x_t, t) = \begin{cases} \mathcal{M} & \text{if } m_t^i \geq (n_t\text{'th highest value of } m_t) \\ x_t^i & \text{otherwise} \end{cases} \tag{7}$$

where $n_t = \lfloor \phi(t, T)L \rfloor$ determines the number of tokens to be masked depending on the time step $t$ and a scheduling function $\phi$. $n_t$ decreases as $t$ increases (more tokens are masked for larger $t$). $m_t^i$ is a generic definition of the masking score of token $i$ at step $t$ and remasking strategies vary in how they compute $m_t^i$. LLaDA applies a linear function for $\phi$ so that an equal amount of $\lfloor \frac{L}{T} \rfloor$ tokens are *not* remasked every step, i.e., $n_t = \lfloor \frac{T-t}{T}L \rfloor$. We investigate various schedules in Appendix B.2.

**Random Remasking (RR):** A simple baseline to construct masking scores is random sampling:

$$m_{t-1}^i \sim \mathcal{U}(0, 1) \qquad \forall_{t,i} : \bar{x}_t^i = \mathcal{M} \tag{8}$$

While random sampling is consistent with the LLaDA training objective (Section 2.1), which masks tokens randomly, this strategy performs poorly in practice (Nie et al., 2025) as it doesn't exploit the structure of natural diffusion trajectories.

**Low-Confidence Remasking (LCR):** Inspired by the annealing tricks of sampling in AR LLMs (Holtzman et al., 2020), LLaDA proposes to utilize the complement of the predicted confidence of the model as the masking scores:

$$m_{t-1}^i = 1 - p_\theta(x_{t-1}^i \mid \bar{x}_t) \qquad \forall_{t,i} : \bar{x}_t^i = \mathcal{M} \tag{9}$$

Empirical evaluations demonstrate that this model-derived low-confidence remasking significantly outperforms random masking in downstream tasks.

**Running Confidence Remasking (RCR):** An important observation is that both remasking strategies above (used in LLaDA) assign masking scores only to predicted tokens in $x_t$ that were masked before. The unmasked tokens remain fixed until the end of the denoising process. In other words, if $x_t^i$ is not remasked by $\gamma(x_t, t)$, it will be unmasked (frozen) in the following steps $x_{<t}^i$. We consider this a crucial limitation, as the predicted tokens, particularly in the early steps, tend to be highly noisy due to the limited structure revealed at that stage. Freezing these noisy tokens until the end of the denoising process makes it more difficult to produce high-quality generation in practice.

To address this, we propose Running Confidence Remasking (RCR). Instead of deciding based solely on the confidence at the current step, we track for each position $i$ the highest confidence it has achieved so far for predicting masked tokens during the denoising process. At each step, we identify the $n_t$ positions whose running maximum confidence is the lowest and remask tokens at these positions. Formally, for token $i$ at step $t$, let $t'$ index any earlier step in the denoising process ($t \leq t' \leq T$), the masking score is defined as:

$$m_{t-1}^i = 1 - \max_{t' \geq t} \left( p_\theta(x_{t'-1}^i \mid \bar{x}_{t'}) \right) \qquad \forall_{t,i} : \bar{x}_t^i = \mathcal{M} \tag{10}$$

Tokens at positions with higher running maximum confidence get lower masking scores (less likely to be remasked), while those that have never reached high confidence remain candidates for remasking.

Empirically, with more structure being revealed along with denoising, earlier steps often yield low-confidence predictions, whereas later steps tend to converge to higher confidence. Under the LCR strategy, early tokens are retained despite their relatively low confidence if they happen to fall within the top-$n_t$ set at that step and can not be refined in future steps. In contrast, as Fig. 2 shows, RCR allows such tokens to be remasked in later steps if tokens at other positions surpass them in running confidence, enabling the model to revise uncertain predictions before producing the final output.

## 3  EXPERIMENTS

Our experiments are designed to address three core questions:

1. How do MDPO and RCR improve generation performance and sampling efficiency, and are their effects complementary?

2. Recognizing over-denoising as a manifestation of the training–inference divide, how can we leverage it to close this gap?

3. What is the impact of different sampling settings during MDPO training?

### 3.1  EXPERIMENTAL SETUP

**Inference:** The performance of MDLMs depends on several inference-time hyperparameter choices. A key distinction lies in whether denoising is performed over the full sequence at once or in blocks. Previous works apply a semi-autoregressive strategy where the sequence is divided into several blocks and generated from left to right. Within each block, tokens are denoised in parallel. We refer to this setting as *semi-AR*, and contrast it with the setting of denoising all tokens in the entire sequence simultaneously (*pure-Diff*). In semi-AR, the same remasking strategies used in pure-Diff apply, but each block denoises fewer tokens with fewer steps. To generate a sequence of length $L$ with block size $B$ in $T$ steps, each block is allocated approximately $\lfloor \frac{T \cdot B}{L} \rfloor$ steps. As shown in Fig. 6, performance varies significantly depending on the choice of $B$ and $T$. For controlled comparisons, we adopt two standard configurations throughout our experiments: pure-Diff ($B = L$) and semi-AR ($B = 128$). Extensive analysis on choices of block size is in Appendix B.3. Though empirically the pure-Diff setting usually underperforms semi-AR, we keep it as an interesting setting in our main evaluation because of the potential speed advantage that pure-Diff possesses at generating long responses.

**Data:** We investigate reasoning tasks with answers that can be verified by static reward functions: (1) MATHEMATICAL REASONING where the model generates solutions to mathematical problems and we verify if the ground truth answers are in the solutions with a robust mathematical expression evaluation system (Kydlíček & Gandenberger, 2025). Note that different math verifiers can lead to changes in evaluation results and we control the experiments by consistently use the same verifier. (2) PLANNING with Countdown (Pan et al., 2025), an easy-to-verify combinatorial arithmetic game in

Table 2: **Model performance on Mathematics and Countdown:** Best and second-best methods in each setting are shaded. For each task we report results on semi-AR (Block Size = 128) and pure-Diff (Block Size = 512) given the generation length of 512. We also compare the performance across multiple choices of denoising steps.

| | MATH500 | | | | | | Countdown | | | | | |
|---|---|---|---|---|---|---|---|---|---|---|---|---|
| Block Size | 128 | | | 512 | | | 128 | | | 512 | | |
| Model / Steps | 64 | 128 | 256 | 64 | 128 | 256 | 64 | 128 | 256 | 64 | 128 | 256 |
| LLaDA-8B-Instruct | 20.0 | 36.2 | 39.4 | 20.0 | 22.6 | 18.2 | 5.5 | 18.8 | 20.7 | 14.8 | 33.6 | 38.7 |
| + SFT (Nie et al., 2025) | 18.0 | 33.8 | 41.8 | 22.2 | 24.6 | 25.4 | 12.3 | 20.7 | 28.6 | 35.5 | 43.1 | 49.6 |
| + diffu-GRPO (Zhao et al., 2025a) | 23.8 | 34.4 | 40.2 | 21.2 | 26.2 | 20.6 | 18.2 | 35.9 | 40.1 | 49.4 | 55.0 | 58.9 |
| + RCR (training-free) | 25.4 | 37.6 | 40.8 | 22.6 | 22.6 | 19.2 | 9.4 | 15.2 | 26.6 | 32.8 | 43.8 | 45.7 |
| + MDPO | 26.2 | 38.6 | 42.8 | 23.4 | 25.2 | 26.2 | 64.8 | 70.7 | 70.7 | 68.4 | 68.8 | 71.1 |
| + MDPO + RCR | 30.4 | 38.4 | 44.2 | 24.8 | 24.0 | 26.4 | 69.9 | 70.7 | 73.4 | 67.2 | 69.5 | 57.4 |

which the model tries to reach target numbers using basic arithmetic operations on a given set (3 or 4) of numbers. Evaluation is on MATH-500 (Lightman et al., 2024), a curated dataset of 500 problems from the MATH dataset (Hendrycks et al., 2021); and the Countdown test set by Zhao et al. (2025a), specifically. For training on the mathematical reasoning task, we utilize (part of) the math data from Huggingface OpenR1 (Hugging Face, 2025) that consists of 93.7k competition-level math problems from NuminaMath 1.5 (LI et al., 2024). For Countdown, we directly use the respective training split released by Pan et al. (2025).

**Model and Training:** We compare our methods to two baseline methods: diffu-GRPO (Zhao et al., 2025a), which is the first integration of policy gradient methods trained on MDLMs, and LLaDA with supervised finetuning (SFT). We (re)train both the baseline methods and MDPO initialized with LLaDA-8B-Instruct (Nie et al., 2025) under the same settings. Training of all methods is conducted under a fixed compute budget using 8 NVIDIA H100 GPUs with 100 weight update steps. We use a batch size of 128 and apply gradient accumulation when memory constraints occur. For both diffu-GRPO and our proposed MDPO, the group size for group-relative advantage estimation is 8, and the number of denoising steps for sampling rollouts is 128. One special setting that will be discussed in detail in Section 3.3 is that we train MDPO with *only* over-denoising samples, which shows better sample efficiency.

### 3.2 Improvements in Generation Performance and Sample Efficiency

We first show the results (see Table 2) of our proposed Masked Diffusion Policy Optimization (MDPO) and training-free Running Confidence Remasking (RCR) compared to the baselines on MATH-500 and Countdown. Across all configurations, both MDPO and RCR individually improve substantially upon the LLaDA initialization, with RCR often achieving performance comparable to MDPO despite requiring no additional training. We remark that RCR, as a training-free method, even outperforms the training baselines in most of the settings for the MATH task. Notably, combining MDPO with RCR consistently yields further gains over either method alone, achieving the best or second-best performance in nearly all settings, which demonstrates that MDPO and RCR are complementary. We further observe that the relative performance gains are more pronounced in settings with fewer inference steps, indicating improved *sampling efficiency*.

**Findings from Countdown:** Furthermore, in Countdown, we observe a surprising behavioral shift: the model trained with MDPO transitions from generating explicit step-by-step reasoning to producing direct answers in a few (<10) steps. This echoes the advantages of diffusion models for latent reasoning discussed by previous work (Zhu et al., 2025). Another interesting observation is that pure-Diff (of LLaDA) substantially outperforms semi-AR on the Countdown task, which is rare on our math task and other tasks evaluated by the LLaDA paper. We hypothesize that this is due to the short expected reasoning traces for Countdown, where semi-AR uses fewer *effective* blocks as well as fewer *effective* denoising steps, limiting its ability to refine predictions. After MDPO training, as the model tends to directly produce answers without explicit reasoning, the performance

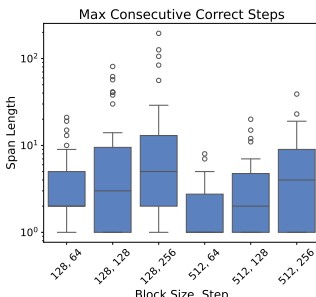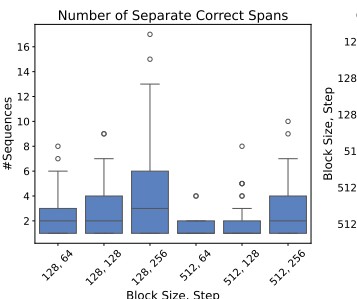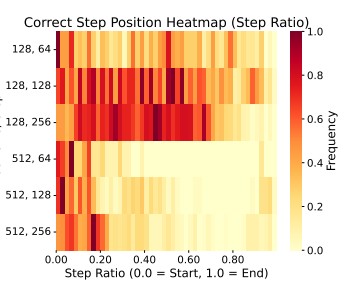

Figure 3: **Analysis on the occurrence of over-denoising samples across multiple inference settings (Block size, Step).** (Left) Maximum length of consecutive correct steps (span) in the denoising trajectories. (Middle) Number of separate correct spans over trajectories. (Right) Heatmap of the relative position of correct steps across the denoising process.

between pure-Diff and semi-AR gets closer. This observation highlights the flexibility of pure-Diff, particularly for tasks that do not require extended reasoning chains.

### 3.3 Over-denoising as An Effective Data Filter

We analyse the over-denoising phenomenon in more detail in Fig. 3 which reveals two key insights. First, the left and middle subfigures show that correct spans are typically short and fragmented, with trajectories often containing multiple separate correct spans instead of steadily accumulating correct steps. Such fragmentation increases the likelihood of losing correct tokens before reaching the final step. Second, the heatmap shows that correct answers often appear surprisingly very early but tend to decay over subsequent steps instead of steadily accumulating.

These findings not only underline the necessity to address the training-inference divide for MDLMs, but also reveal that over-denoising provides highly informative signals of intermediate steps for improving MDLMs. Inspired by this, we train MDPO with only over-denoising samples instead of all data so that MDPO can learn from these intermediate signals more efficiently. Specifically, we first use the initialized model to run inference on the whole dataset to identify over-denoising samples and then train only on this subset, which constitutes roughly 10% of the original data. The comparison between MDPO-all-data and MDPO in Fig. 4 shows that the model trained on only over-denoising excels in most settings, highlighting the effectiveness of over-denoising as a data filter for MDPO.

### 3.4 Impact of Sampling Settings on MDPO

The heatmap in Fig. 3 reveals that semi-AR tends to distribute correctness more evenly but with greater fragmentation, while pure-Diff produces an early burst of correctness that decays sharply and rarely recovers. This raises a question of how different sampling settings during the rollout collection of RL affect the final performance. To investigate this, we compare MDPO variants trained on (i) pure-Diff rollouts only, (ii) semi-AR rollouts only, and (iii) an even mixture of both. Fig. 4 shows that training on a single mode yields the largest improvement in the evaluation setting of the respective mode, but often at the cost of performance in the other modes. The mixture strategy used in our main MDPO setting achieves a more balanced performance, matching or exceeding the best single-mode results in several configurations. This suggests that mixed sampling allows the policy to learn denoising behaviors that generalize across all inference strategies.

## 4 Related Work

Diffusion probabilistic models (Sohl-Dickstein et al., 2015; Ho et al., 2020) have become the de facto standard for generative modeling of continuous signals, including images, videos, 3D shapes, and robotic trajectories. Building on early efforts in discrete diffusion (Austin et al., 2021; Hoogeboom et al., 2021), several works have adapted diffusion models for text generation. Diffusion-LM (Li et al., 2022) and Self-Conditioned Embedding Diffusion (Strudel et al., 2022) have pioneered embedding-based diffusion approaches. Liu et al. (2025) introduces a novel framework that separates the

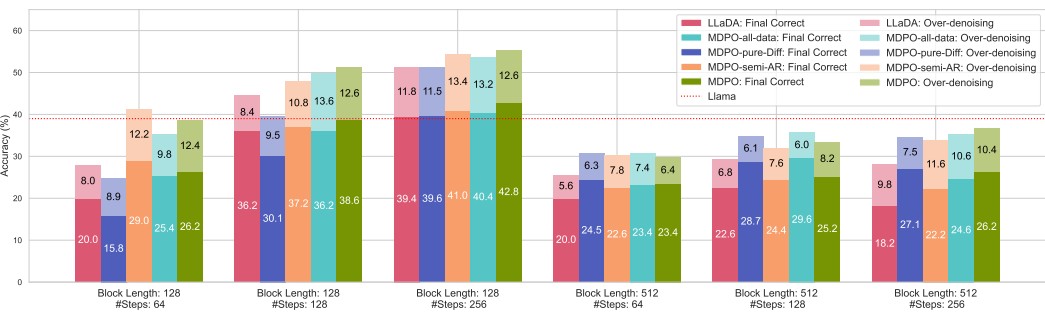

Figure 4: **Comparison of MDPO variants.** We report the final accuracy and proportion of over-denoising cases for all models. MDPO-all-data is trained on all data samples, whereas MDPO is trained on *only* over-denoising data, both with rollouts sampled from a mixture of semi-AR and pure-Diff. MDPO-pure-Diff and MDPO-semi-AR represent models that are trained on rollouts sampled from *only* pure-Diff, and semi-AR, respectively.

generation process into two models: a planner and a denoiser. Meanwhile, masked diffusion (Austin et al., 2021; Sahoo et al., 2024; Ou et al., 2025) has been established as a special case of discrete diffusion, with recent research significantly scaling their capabilities. Notably, DiffuLLaMA (Gong et al., 2025) initializes masked diffusion language models using pre-trained LLaMA weights. LLaDA (Nie et al., 2025) and Dream (Ye et al., 2025) scale MDLMs by pre-training from scratch to the billion-parameter regime, achieving performance comparable to similarly sized AR LLMs.

Training diffusion models with RL has been primarily explored in image diffusion models (Black et al., 2024; Fan et al., 2023) on non-differentiable objectives such as human-perceived image quality. Represented by Group Relative Policy Optimization (GRPO) (Shao et al., 2024), RL has emerged as a promising tool to optimize sparse and task-specific reward signals for language models (Guo et al., 2025; Yu et al., 2025b; Luo et al., 2025b;a; Hu et al., 2025; Zeng et al., 2025), especially for post-training. These RL algorithms assume sequential token-by-token generation, a key property of AR models. However, MDLMs differ fundamentally: they denoise entire sequences from corrupted inputs, making AR-specific RL inapplicable. Recent attempts Zhao et al. (2025a;b) adapt GRPO to MDLMs with inserted partial ground-truth reasoning traces to improve sample efficiency, meanwhile concurrent work Wang et al. (2025b) also observes the over-denoising phenomenon and adds entropy-based rewards to optimize MDLMs with RL. In contrast, we introduce MDPO, a policy optimization method designed for MDLMs to learn effective denoising trajectories by optimizing the iterative denoising process as a multi-step decision-making problem. To our knowledge, MDPO is the first RL method explicitly designed to optimize MDLMs denoising trajectories as multi-step decision-making.

Kim et al. (2025); Wang et al. (2025a) show that though MDLMs face inherently harder training challenges due to solving computationally intractable infilling tasks, using adaptive decoding strategies effectively alleviates these. Following this, Block Diffusion (Arriola et al., 2025) models sequences in autoregressive blocks with intra-block diffusion, concurrent work Li et al. (2025) observes the same over-denoising phenomenon and applies a confidence-based early-commit strategy for faster denoising. Our proposed Running Confidence Remasking enables more flexible decoding to substantially enhance the denoising quality by enabling iterative revision of early-predicted 'noisy' tokens.

## 5 DISCUSSION & CONCLUSION

This paper tackles the training–inference divide in Masked Diffusion Language Models (MDLMs) by proposing Masked Diffusion Policy Optimization (MDPO) and Running Confidence Remasking (RCR). MDPO uses reinforcement learning to optimize denoising trajectories with intermediate rewards, while RCR flexibly revisits earlier noisy predictions to reduce error propagation. One limitation of our work is that only verifiable tasks with static reward functions are investigated. We leave the application of MDPO to general tasks using modern validation concepts such as LLM-as-a-judge for intermediate rewards as future work.

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

# Appendix

## Table of Contents

## A  EXPERIMENTAL SETTINGS

### A.1  ADVANTAGE ESTIMATION

In Eq. (6), we define the score of each generation as

$$s_{t,k} = r(x_{t,k}) - r(x_{t+1,k}) + \frac{1}{t}\sum_{j=0}^{t-1} r(x_{j,k}) \tag{11}$$

to compute the advantage. The intuition behind this is the first part of the definition $r(x_{t,k}) - r(x_{t+1,k})$ incentivizes immediate improvement over the previous, noisier prediction and the second part $\frac{1}{t}\sum_{j=0}^{t-1} r(x_{j,k})$ encourages predictions that steer the trajectory toward cleaner states that score well overall. We highlight that the design of advantage estimation plays a pivotal role in determining the final performance, since poorly structured estimators can be easily hacked, allowing the model to overfit to superficial reward patterns rather than acquiring the intended behavior.

### A.2  KL DIVERGENCE APPROXIMATION

We approximate the KL divergence between the trained model and the reference model $\mathbb{D}_{KL}\left[p_\theta || p_{\text{ref}}\right]$ in Eq. (4) following Schulman (2020). Define $d = \frac{p_{\text{ref}}(x)}{p_\theta(x)}$, three estimators are defined in Schulman (2020):

$$k1 = -\log d \qquad k2 = \frac{(\log d)^2}{2} \qquad k_3 = (d-1) - \log d \tag{12}$$

where $k1$ is an unbiased but high-variance estimator, as it is negative for half of the samples, while the KL divergence should always be positive. Using such an estimator inside Eq. (4) where the KL divergence is deducted from the clipped advantage may artificially reward models for 'negative divergences', producing biased optimization dynamics. Therefore, we seek an estimator that is always non-negative, for robustness and theoretical consistency with the definition of KL. $k2$ and $k3$ are both non-negative, and $k2$ is more biased than $k3$, according to Schulman (2020). In our preliminary experiments, we found small differences in the performance between using $k2$ and $k3$. Therefore, we choose $k2$ as the final KL approximation in our current implementation for the simplicity of computation.

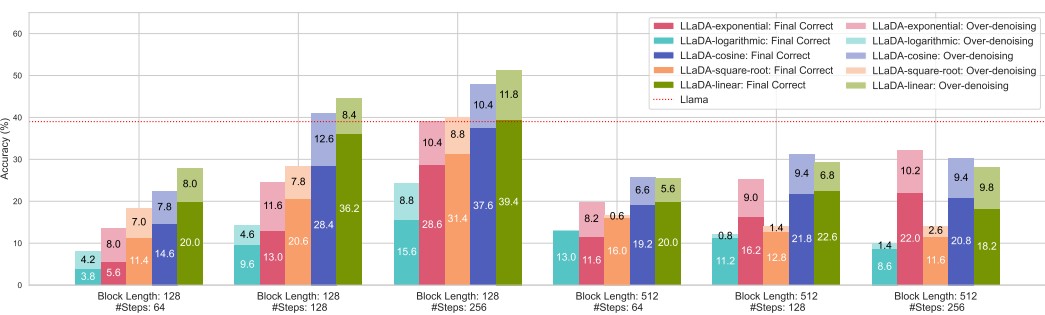

Figure 5: Comparison of different re-masking strategies introduced by MaskGIT. The performance of LLaDA with changing re-masking strategies is not consistent with MaskGIT, a image generation model.

### A.3 HYPERPARAMETER SETTING & OTHER TECHNICAL DETAILS

Here we list other technical details that are necessary to reproduce our experimental results. We set the clipping parameter $\epsilon = 0.2$ in the objective in Eq. (5) and the KL divergence scaling factor $\beta = 0.08$ in Eq. (4). In our experiments, we observe that a larger $\beta$ prevents the model from overfitting to the training data, given that we are supervising the intermediate trajectories and the model receives multiple supervisions for each data sample. Another implementation choice we make to avoid overfitting is to reduce the number of intermediate denoising steps used to create the gradients. Despite the expectation is over all the denoising steps as defined in Eq. (4), we sort the denoising steps according to their absolute advantage values and pick the largest $c$ steps. This not only prevents the model from overfitting by reducing the number of the same data samples are trained on, but also reduces the computation time. $c$ is usually set to $8$ in our experiments. The intuition behind sorting advantages with absolute values is to supervise the model with both good and bad denoising. Learning rate is set to 7e-7.

In Section 3.3, we find that using only over-denoising samples is more data-efficient and produces better results. Although this leads to overhead in pre-computing the over-denoising samples before training, it is still computationally efficient in practice. The reason is that for each inference setting, we only need to infer over the whole training set once and use the over-denoising data samples repetitively for sampling rollouts in the online RL in one training or over multiple training sessions. As the computation cost of inference is much less than completing forward and backward propagation in training, the overhead created by inference is negligible compared to the computation cost for training on all data.

## B  MODEL ANALYSIS

### B.1  TRAINING-INFERENCE DISCREPANCY

In Section 2.4, we introduce the two remasking strategies from LLaDA, random remasking and low-confidence remasking (LCR). As the tokens are also randomly masked in pre-training, and LCR is usually used as the de-facto standard for inference (Nie et al., 2025), analyzing the difference between random remasking and low-confidence remasking provides us a closer look at the training-inference divide. Table 3 shows concrete examples to demonstrate the difference between random masking and LCR.

We observe that LCR consistently produces more coherent and complete masked intermediate completions compared to random remasking at the same masking ratio. At high masking ratios (e.g., 0.75 at Step 16), random remasking often fails to preserve the problem structure, leading to fragmented or nonsensical outputs, whereas LCR retains recognizable mathematical expressions and approaches the ground truth earlier. As the masking ratio decreases (Steps 32 and 48), the gap narrows, while LCR still maintains a higher degree of structural integrity.

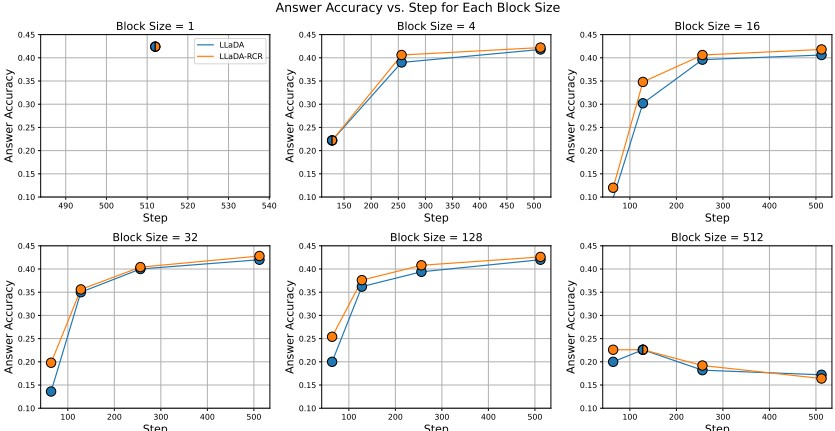

Figure 6: Interpolation of block sizes and number of time steps to investigate the effect of these two factors on the inference performance. Generation length is fixed to 512.

## B.2 REMASKING SCHEDULING FUNCTION

The remasking scheduling function $\phi$ introduced in Section 2.4 defines how many tokens should be masked at every timestep, which can also influence the generation performance. Though the choices of $\phi$ is not discussed in the LLaDA paper, multiple design choices of the $\phi$ are studied in MaskGIT (Chang et al., 2022) for image generation tasks. We list the definitions of these functions as following

$$
\phi(t, T) = \begin{cases}
\dfrac{e - e^{\frac{t}{T}}}{e - 1}, & \text{Exponential} \\[2ex]
1 - \left(\dfrac{t}{T}\right)^3, & \text{Cubic} \\[2ex]
1 - \left(\dfrac{t}{T}\right)^2, & \text{Square} \\[2ex]
\dfrac{1 + \cos\left(\pi \frac{t}{T}\right)}{2}, & \text{Cosine} \\[2ex]
1 - \dfrac{t}{T}, & \text{Linear} \\[2ex]
1 - \sqrt{\dfrac{t}{T}}, & \text{Square Root} \\[2ex]
1 - \dfrac{\ln\left(1 + \frac{t}{T}\right)}{\ln(2)}, & \text{Logarithmic}
\end{cases}
\tag{13}
$$

Out of these functions, LLaDA uses the linear strategy that the same amount of tokens are *not* remasked at each step. We conduct evaluation with the representative ones: exponential (exp), logarithmic (log), cosine, square root, and linear, on the MATH-500 task. Results in Fig. 5 show that the linear strategy works the best, which is not consistent with MaskGIT on image synthesis, where the concave functions like cosine that follows a less-to-more generation procedure works better.

## B.3 BLOCK SIZE IN SEMI-AR

As discussed in Section 3.1, a key factor that affects the performance of MDLMs is the block size. When block size $B = L$ where $L$ is the generation length, the model denoises all the tokens simultaneously which we refer to as *pure-Diff*. And we refer to $B < L$ as *semi-AR* where the sequence is divided into several blocks and generated from left to right. Within each block, tokens are denoised in parallel.

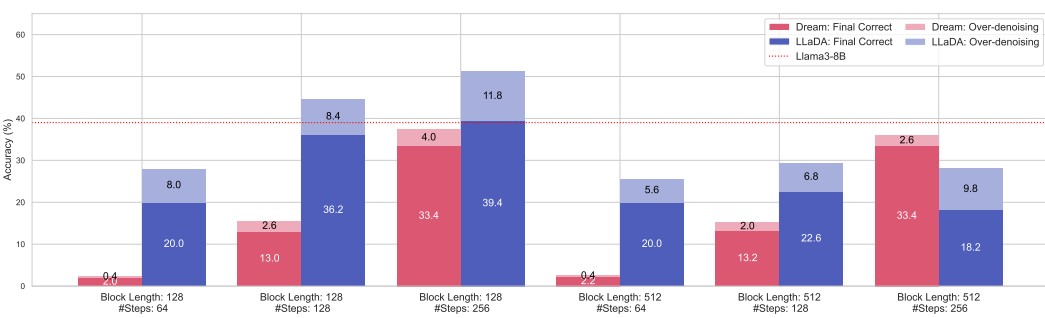

Figure 7: Comparison of LLaDA and Dream on MATH. Results indicate that the performance of Dream is highly dependent on the number of denoising steps, while LLaDA benefits more from the semi-AR setting (with block length 128).

To better understand the interplay between block size and the number of denoising steps, we conduct experiments with LLaDA-Instruct on the MATH-500 benchmark, varying $B \in 1, 4, 16, 32, 128, 512$ and the number of steps $T \in 32, 64, 128, 256, 512$. The results in Fig. 6 reveal several trends: (1) Small block sizes (e.g., $B = 4$) show slower initial gains but benefit more from increasing $T$, indicating that finer-grained autoregression allows iterative refinement over more steps; (2) Medium block sizes (e.g., $B = 16, 32, 128$) achieve the highest accuracy at moderate $T$, suggesting a favorable balance between parallelism and left-to-right dependency modeling; (3) pure-Diff (i.e., $B = 512$) consistently underperforms, and accuracy even drops at higher $T$, implying that excessive parallel denoising hinders the model's ability to progressively refine solutions on math tasks. However, as discussed earlier following Table 2, pure-Diff is more flexible in dealing with tasks that require answers with varied length, e.g., Countdown. Therefore, improving the performance of pure-Diff is as well crucial to the wider application of masked diffusion language models.

### B.4 LLaDA AND DREAM

As an extended analysis, we choose another pre-trained MDLM Dream 7B (Ye et al., 2025) to compare with LLaDA. Note that Dream 7B builds on a different algorithm to LLaDA and initializes from an auto-regressive (AR) checkpoint Qwen2.5 7B (Yang et al., 2024). Though the algorithm still follows the absorbing discrete diffusion framework to transit tokens to and from the predefined absorbing state (masking token), Dream differs in predicting the next token and filling the next token with the prediction if the next token is masked. This technical difference originates from the fact that Dream is adapted from an AR model which is trained on next token prediction. We observe a clear behavioral difference between the pre-trained from scratch LLaDA and AR-adapted Dream, on MATH task shown in Fig. 7.

The first observation is that the performance of Dream is highly dependent on the denoising steps, but not the block length. In the semi-AR setting (Block Length of 128 in Fig. 7), which substantially improves the performance of LLaDA, Dream achieves downstream performance similar to the pure-Diff setting. We hypothesize that this arises because the model is adapted from an AR model, where earlier tokens consistently receive higher confidence. Consequently, pure-Diff effectively converges toward semi-AR, since tokens in early positions typically obtain higher confidence scores and are rarely remasked. Evidence for this convergence is shown in Fig. 8, where Dream predominantly assigns higher confidence to earlier masked tokens.

Our second observation is that Dream exhibits a smaller degree of over-denoising compared to LLaDA. We hypothesize that this difference also stems from its adaptation from AR models. While we see over-denoising as a symptom of the training–inference divide inherent in masked diffusion, Dream is pre-trained on a mixture of objectives, including both next-token prediction and masked diffusion. In particular, the pre-training of Dream on the masked diffusion objective consumes 580 billion tokens, whereas the original initialization Qwen 2.5 pre-training covers 18 trillion tokens. We argue that this imbalance in the scale of pre-training tokens between the two stages reduces the extent to which Dream experiences the training–inference divide, making its behavior more similar to that of an AR model. As a result, the discrepancy is less pronounced in Dream, and applying our proposed methods might yield only marginal improvements.

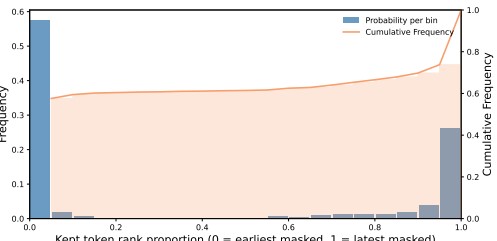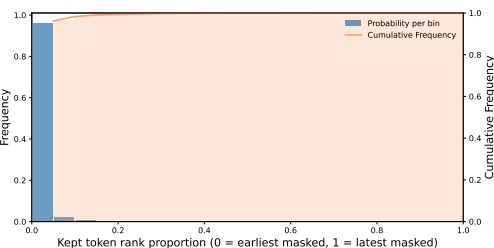

Figure 8: Quantitative analysis of predicted and 'kept' (not remasked) tokens in LLaDA and Dream. We apply confidence-based remasking with pure-Diff to both models and plot the distribution of retained tokens by their rank among masked positions at respective denoising steps across trajectories. The results show that Dream tends to assign higher confidence to earlier masked tokens, resembling AR models, whereas LLaDA exhibits a two-sided pattern, giving higher confidence to both early and late masked tokens.

## C   ADDITIONAL DISCUSSION ON EXPERIMENTAL RESULTS

### C.1   WHY OVER-DENOISING IS AN EFFECTIVE DATA FILTER?

In Section 3.3, we demonstrate that training solely on over-denoising samples leads to significant performance improvements. Our intuition for this advantage is twofold. (1) Over-denoising samples yield more informative gradients because predictions often flip back and forth between correct and incorrect answers. This dynamic allows the model to learn from denoising steps that steer predictions in both directions, thereby stabilizing its outputs over time. (2) Advantages are mostly non-zero if we use over-denoising samples for online sampling. As we only use the binary rewards in our settings, advantages are either 0 or 1 for any denoising step. For denoising steps that fail to produce correct answers, rewards are uniformly zero, and exploration collapses if all rollouts in a group obtain zero rewards. This is precisely the well-known zero-advantage problem in training LLMs with RL (Yu et al., 2025a; Zhao et al., 2025b). To solve this issue, Zhao et al. (2025b) proposed a novel inpainting method for MDLMs to steer exploration toward promising trajectory spaces, thereby mitigating the zero-advantage issue and improving sample efficiency. We view open-denoising filtering as an alternative approach to addressing the same problem.

## D   THE USE OF LARGE LANGUAGE MODELS

Large language models are used in our early-stage writing for wording and grammar checking, as well as for searching for missing literature. LLMs are not involved in the later iterations of the paper writing. Therefore, we do not consider LLMs as significant contributors to this paper.

Table 3: Comparison of random remasking and confidence-based remasking. We show the model completions at selected denoising steps (16, 32, 48) given a prompt, with the remasked tokens ([M]) by random remasking and LCR with a mask ratio corresponding to the respective step.

| Prompt | Define $p = \sum_{k=1}^{\infty} \frac{1}{k^2}$ and $q = \sum_{k=1}^{\infty} \frac{1}{k^3}$. Find a way to write $\sum_{j=1}^{\infty} \sum_{k=1}^{\infty} \frac{1}{(jk)^3}$ in terms of $p$ and $q$. |
|---|---|
| Ground Truth | \boxed{p - q} |
| *Step 16:* Completion | We can write\n \[\sum_{j = 1}^\infty \sum_k = 1^\infty \frac1(j + k)^3 = \sum_1 = 1^ \infty \frac1k^^ = \boxed{pboxed{p - q}.\] |
| *Remasking Ratio 0.75* | |
|    Random | [M][M][M][M]\[\[M][M][M][M] [M][M][M][M]sum[M][M][M][M]1[M]infty \[M][M][M][M][M][M][M][M][M][M] = \[M][M][M][M][M]1[M][M][M][M]{[M]}{k^[M][M][M][M][M][M][M][M][M][M][M][M] |
|    LCR | We can write\n \[\[M]_{[M] = 1[M]infty[M][M][M][M][M][M][M][M][M][M][M][M][M][M][M][M][M][M][M][M][M][M][M][M][M][M][M][M][M][M][M][M][M][M][M][M][M][M][M][M][M][M][M][M] - q}.\] |
| *Step 32:* Completion | We can write\n\[\sum_{j = 1}^\infty \sum_{k = 1}^\infty \frac{1}{(j + k)^3} = \sum_{j = 1}^\infty \frac{1}{k^^} = \boxed \boxed{p - q}.\] |
| *Remasking Ratio 0.5* | |
|    Random | [M] can write[M]\[\[M]_{j[M] 1[M]infty \sum[M][M][M][M]}^\[M] \[M][M]1[M][M] +[M][M]3[M] =[M][M]_{j =[M]1}^\infty\frac[M]1}{[M][M][M][M] =[M][M]\[M][M][M][M]}.[M][M][M] |
|    LCR | We can write\n\[\sum_{j = 1}^\infty \sum_{k = 1}^\infty \frac{1[M][M][M][M][M][M][M][M][M][M][M][M][M][M][M][M][M][M][M][M][M][M][M][M][M][M][M][M][M][M][M][M] - q}.\] |
| *Step 48:* Completion | We can write \n\[\sum_{j = 1}^\infty \sum_{k = 1}^\infty \frac{1}{(j + k)^3} = \sum_{j = 1}^\infty \frac{1}{j^3} = \boxed{p}{{p - q}.\] |
| *Remasking Ratio 0.25* | |
|    Random | We[M] write [M]sum[M]j[M] 1}^\infty[M]sum_{k =[M][M][M]infty \[M]{[M]}{(j + k[M]3} =[M]sum_{[M] = 1}^\infty \[M]1{j^3[M] =[M][M]{p}{{p - q}.\] |
|    LCR | We can write\n\[\sum_{j = 1}^\infty \sum_{k = 1}^\infty \frac{1}{(j + k)^3} = \sum_{[M] = 1}^\infty[M][M][M][M][M][M][M][M][M][M][M][M][M][M][M] - q}.\] |
| *Step 64*: Final Completion | We can write\n\[\sum_{j = 1}^\infty \sum_{k = 1}^\infty \frac{1}{(j + k)^3} = \sum_{j = 1}^\infty \frac{1}{j^3} = \boxed{2p - q}.\] |

