# OpenReview forum: "MDPO: Overcoming the Training-Inference Divide of Masked Diffusion Language Models"
_ICLR.cc/2026/Conference — ICLR 2026 Conference Withdrawn Submission_

### Official Review · Reviewer_hVw8 · 2025-10-31

**Soundness:** 2
**Presentation:** 2
**Contribution:** 2
**Rating:** 2
**Confidence:** 4

**Summary:**

This paper introduces the Masked Diffusion Policy Optimization (MDPO) algorithm, a finetuning method designed to resolve training-inference inconsistencies in LLaDA-family models. These inconsistencies arise from the divergence between unbiased stochastic estimation used in training masked diffusion models and the biased, heuristic-based inference, including the confidence-based remasking strategy employed during inference.

MDPO reframes the fine-tuning process as a trajectory optimization problem, drawing inspiration from reinforcement learning. Instead of optimizing timestep-by-timestep, MDPO fine-tunes models at the trajectory level by sampling complete generation trajectories during each iteration.

Additionally, the author proposes an enhanced remasking strategy called Running Confidence Remasking (RCR). RCR remasks tokens by incorporating historical predictions from along the generation trajectory.

The proposed methods yield good improvements on the MATH500 and Countdown benchmarks. Moreover, the authors also provide detailed ablation analysis of block sizes and total inference steps.

**Strengths:**

The proposed methods yield strong improvements on the MATH500 and Countdown benchmarks. In addition, the authors also provide detailed ablation analysis of the effect of different block sizes and total inference steps.

**Weaknesses:**

While the proposed methods show improvement, the authors overstate the training-inference discrepancy. The true challenge is the high cost of sampling full generation trajectories during training. Unbiased stochastic estimation, by sampling one trajectory step, offers a good trade-off but increases variance. This variance propagates during inference; confidence-based sampling could reduce it but introduces bias. These core issues are not clearly discussed.

Finetuning an MDLM, trained with unbiased stochastic estimation, at the trajectory level is a legitimate approach to reduce sampling variance. PPO, adopted in MDPO, is also an effective measure for variance reduction, though at the cost of increased computational expense. MDPO, in particular, exacerbates this by sampling multiple trajectories per iteration. Moreover, fine-tuning masked diffusion models via RL is also an active direction in the literature (e.g., [1]).

In addition to the major issue about presentation and motivation, here are a few other issues, addressing which could help to improve the quality of the paper:

1. Lack of a more comprehensive evaluation on downstream tasks. This paper only evaluates on MATH500 and Countdown, while the standard evaluation suite in the literature also includes GSM8K and Sudoku.

2. A complexity analysis of MDPO will be very helpful, including time complexity, actual training time, etc.

3. It’d be fair to compare against other methods under the similar compute budget, or at least report the compute required for each method.

4. Inaccurate claims: Line#071 “at inference time, MDLMs follow a model-dependent, confidence guided remasking schedule”. I think this is just a specific remasking strategy used by some models (e.g., LLaDA), which is one auxiliary and optional step in inference, while other models (e.g., MD4 [2], Dream-8B [3]) still use unbiased sampling in their original implementations.

5. Inconsistent claims: while Line#071 states that remasking is the key discrepancy between training and inference, Line#016 implies that the sequence/trajectory structure being ignored is the key discrepancy.

6. The authors' claim in Section 3.3, line 407, regarding MDPO's initial inference on the entire dataset to gather over-denoised samples, raises concerns. If the initialized model is pre-trained LLaDA, this creates an unfair comparison against MDPO-all-data. The process involves both increased computational cost and manual identification of failed samples, making it unsurprising that targeting these with a reward function improves performance.

Minor issues:
1. The paper's clarity is hampered by the delayed explanation of the role of MDPO. It's not until Section 2.2, Line 157, that the authors clarify MDPO is a fine-tuning algorithm, rather than a pre-training algorithm as initially implied (e.g., Line#31). If MDPO is also suitable for pre-training, the authors should provide relevant evaluation and evidence.

[1] https://arxiv.org/pdf/2410.13643
[2] https://arxiv.org/abs/2406.04329
[3] https://arxiv.org/abs/2508.15487

**Questions:**

1. The authors claim that MDPO can learn effective discrete diffusion trajectories without direct supervision. However, in equation (4),(5) and (6), the learning objective still relies on reward r(x_{t}), which is evaluated via an evaluation model (Line#185). Please can you elaborate?

2. Effect of different group sizes |G|. Please can you provide an ablation analysis of different group sizes |G|.

3. When you collect the over-denoised samples, are they just the final incorrect samples or do they also include the partial trajectories with correct tokens?

4. What’s the task reported in Fig 4?

5. The title of Section 3.2 should be “sampling efficiency”?

---

### Official Review · Reviewer_s9wR · 2025-11-01

**Soundness:** 2
**Presentation:** 2
**Contribution:** 1
**Rating:** 2
**Confidence:** 4

**Summary:**

This paper introduces a policy optimization and inference-time decoding approach for masked diffusion models. In particular, the policy optimization adopts a GRPO-like objective to align the stepwise policy against the reward that evaluates the per-step prediction of the clean sequence. At inference time, the token as the position with the running lowest confidence in history is remasked. Experiments on math and countdown shows improvement brought by the approach in terms of enhancing the accuracy.

**Strengths:**

1. The identification of the over-denoising phenomenon seems interesting. The paper proposes a policy optimization approach to mitigate the problem and a corresponding decoding method that can further enhance the performance.

2. The method proposed is easy to follow and the flow of the paper is clean.

**Weaknesses:**

1. The experiment is only performed on LLaDA-Instruct on two tasks which is not comprehensive enough.

2. The proposed policy optimization algorithm lacks theoretical insights. It is unclear how this objective can optimize the policy towards a more favorable one and what is the relationship between the reward model and the optimized policy.

3. Similarly, the running confidence remasking (RCR) is also proposed in a rather heuristic way. It is unclear how RCR is related to enhance performance and why it is compatible with the RL training.

**Questions:**

1. How does the method scale to other important benchmarks such as GSM8K (and potentially coding benchmarks like humaneval/MBPP [1])?

2. Can the authors provide more insights on why the policy optimization is effective? In particular, the per-step reward seems like pretty vague when the diffusion step is large since the model can not really give a good answer at that time. Why would training on those steps also help?

3. How does RCR compared with some other advanced decoding approaches such as [2]?

[1] Gong et al. DiffuCoder: Understanding and Improving Masked Diffusion Models for Code Generation.

[2] Kim et al. Train for the Worst, Plan for the Best: Understanding Token Ordering in Masked Diffusions.

---

### Official Review · Reviewer_kAR7 · 2025-11-01

**Soundness:** 3
**Presentation:** 3
**Contribution:** 3
**Rating:** 4
**Confidence:** 3

**Summary:**

This work tackles what they term the training-inference divide, a mismatch in how discrete LMs are trained where tokens are masked at random and how discrete LMs are used at inference time where the denoising process selects tokens to unmask at a given step, resulting in some sort of non-random structure depending on multiple factors (the model, the denoising process, the prompt used, etc). This paper proposes Masked Diffusion Policy Optimization (MDPO), a RL based method aimed at refining the denoising process and RCR, a deterministic version of LLaDa style confidence based de-noising process that keeps track of confidences over a horizon instead of at the current denoising step. This work then evaluates their contributions on MATH-500 and Countdown using LLaDa-8B Instruct as the base model and conduct some additional analysis on "over-denoising", a phenomenon where the model produces a correct answer during the denoising process but where this correct partial answer is refined into an incorrect answer by the end of the denoising process.

**Strengths:**

The paper is well written and presented, the general set-up seems reasonable and I believe the authors claims are generally supported by their experiments. The topic of de-noising and how it relates to down-stream tasks w.r.t performance and efficiency is a useful area of research for diffusion models, and this work poses a timely addition.

**Weaknesses:**

The main concern I have with this work is that only two tasks are evaluated. While their current evaluation seems reasonable to me, having only two tasks and being limited to only verifiable tasks really limits the scope of this work. The authors do partially address this in the work, but in my opinion I believe there needs to be some sort of analysis on tasks that are not easily verifiable to quantify possible error modes/limitations of MDPO as a function of the "difficulty to verify" the task, or to attempt more open ended tasks. For example for RCR I would like to see some eval on generation for example, something like MAUVE or Gen.PPL measures on a dataset (OWT for example) similar to ReMDM.

Expanding the evaluation (ideally by including more tasks, above is an example but there are multiple such tasks that I think would be applicable) would greatly help the strength of this work.

**Questions:**

I would like some more analysis on how many gradient updates are needed (for all relevant methods). I understand that in this work a fixed compute budget was used for all methods, I'm specifically asking how does the performance of different methods compare under a few different compute budgets.

I would also like some more comparisons to other Re-Making work, particularly comparisons to contemporary relevant like ReMDM and predictor-corrector approaches like Discrete Flow Matching. How do the methods form these works compare to this work, (and if possible I'd also like to see this work evaluated against them).

---

### Official Review · Reviewer_W6sn · 2025-11-10

**Soundness:** 2
**Presentation:** 3
**Contribution:** 3
**Rating:** 4
**Confidence:** 4

**Summary:**

This paper proposes MDPO and RCR. MDPO is a RL post training method for masked diffusion models that combines a novel reward function and PPO and GRPO. The reward function explicitly encourages improvement in each denoising step. RCR is a refinement strategy that builds on the remasking technique from LLaDA, allowing tokens to be remasked in steps after they have been unmasked.

**Strengths:**

* The reward function definition that encourages improvement in each denoising steps is interesting (and novel I believe). However, a thorough ablation is missing to demonstrate the usefulness of including such a term - how does it compare to just optimizing the total reward? I will consider increasing the score if this is included in the rebuttal phase.

* The observation made about "remasking" strategy in LLaDA is interesting - such remasking only happens in the same step where the same token is unmasked. This clearly has a disadvantage. The proposed fix, allowing tokens to be remasked in later steps, is sensible and leads to empirical gain.

**Weaknesses:**

* The authors seem to interchangeably use the terms "masked diffusion" and "diffusion language models" which are related but not exactly same concepts. Masked diffusion (notably used for text but not only text, e.g., D3PM and MD4 apply to images too) did not use confidence-based inference nor remasking. The confidence-based inference and remasking was first introduced in the language modeling context by LLaDA. Therefore, I find the name of the method (MDPO) as well as many statements such as "MDLMs follow a model-dependent, confidence-guided remasking schedule that progressively reveals the structure of generated sequence" inaccurate. As a reader relatively familiar with masked diffusion models, I also got confused by "correct immediate solutions are 'refined' into wrong final solutions" because I know no refining of solutions happens in such models. I realized later that this was speaking of LLaDA like models. The authors are encouraged to revise the paper and correct relevant statements.

* It's unclear how the authors deal with masks in intermediate steps - how do you verify the answer with [mask] inside the generation? I find statements like "A key advantage of masked diffusion language models .... is that they yield complete text generations at every inference step" unclear. In what sense you call them "complete"?

* It is unclear how MDPO alone (without RCR) improves upon methods like diffu-GRPO since they look very similar. The paper should include a discussion on this.

**Questions:**

Please see questions in above sections (I will consider increasing the scores if they are sufficiently addressed).

---

### Author Response · Authors · 2025-12-04
**Thank you for the valuable comments**

We sincerely thank all reviewers for their hard work and constructive feedback. We highly value the insights and suggestions provided, and we will incorporate them to further improve our work in future revisions. The changes we are making are summarized below:

1. Ablation on different parts of the reward function mentioned by **reviewer W6sn**. We agree that this is an important experiment to understand the effectiveness of the reward function better. We will update the results in our next version of the paper.

2. Expanding the evaluation on GSM8k and coding benchmarks, mentioned by **reviewers kAR7 and s9wR**.

3. Ablation study on different group sizes, mentioned by **reviewer hVw8**.

4. Comparing RCR to previous works such as ReMDM, mentioned by **reviewer kAR7**.

5. Better phrasing to avoid confusing the readers about the terms "masked diffusion" and "diffusion language models", mentioned by **reviewers W6sn and hVw8**. Note that MDPO is introduced specifically based on the phenomenon and features of masked diffusion language models, such as LLaDA. We didn’t mean to generalize to masked diffusion, though it is not clearly discussed in the paper. We will iterate the phrasing in our next revisions.

After careful consideration, we have decided to withdraw the paper at this time given the amount of experiments we need to do. We are truly grateful for your time, expertise, and guidance.

---

### Note · Authors · 2026-03-27

I have read and agree with the venue's withdrawal policy on behalf of myself and my co-authors.

---

### Meta-Review · Area_Chair_CWGT · 2026-01-05

**Summary:**

This paper proposed two methods to fill the training/inference gap for masked diffusion LMs. Reviewers think that it has some interesting ideas, but at the same time fall short on rigorous evaluations as well as clear presentations. The authors have agreed and decided to withdraw this paper (thought haven't actually clicked the button). The AC respects the authors decision.

**Reviewer Concerns:**

No rebuttal provided.

**Reviewer Scores:**

No rebuttal provided.

---

### Decision · Program_Chairs · 2026-01-26

Reject